# Lives of Skin Lesions in Monkeypox: Histomorphological, Immunohistochemical, and Clinical Correlations in a Small Case Series

**DOI:** 10.3390/v15081748

**Published:** 2023-08-15

**Authors:** Paul Schmidle, Sonja Leson, Ulrike Wieland, Almut Böer-Auer, Dieter Metze, Stephan A. Braun

**Affiliations:** 1Department of Dermatology, University Hospital Muenster, University of Muenster, 48149 Muenster, Germany; sonja.leson@ukmuenster.de (S.L.); boer@dermatologikum.de (A.B.-A.); dieter.metze@ukmuenster.de (D.M.); stephanalexander.braun@ukmuenster.de (S.A.B.); 2Institute of Virology, Faculty of Medicine and University Hospital Cologne, University of Cologne, 50931 Cologne, Germany; ulrike.wieland@uk-koeln.de; 3Dermatologikum Hamburg, 20354 Hamburg, Germany; 4Department of Dermatology, Medical Faculty, Heinrich-Heine University, 40225 Duesseldorf, Germany

**Keywords:** histology, pathology, sexually transmitted infection/diseases, immunohistochemistry, electron microscopy, vasculopathy

## Abstract

Monkeypox (mpox), a former rare viral zoonosis, has increasingly made it into the public eye since the major outbreak that started in May 2022. Mpox presents with skin lesions that change over time and go through different stages (macular, papular, pustular, and early and late ulceration). In this study, we evaluated skin biopsies of all stages. Therefore, five biopsies from four patients were analyzed histologically, immunohistochemically with anti-*Vaccinia virus* antibodies, and electron-microscopically. Notably, the early macular stage only showed subtle viropathic changes; it did not express of Orthopoxvirus proteins in immunohistochemistry and therefore can easily be missed histologically. In later stages, immunohistochemistry with anti-*Vaccinia virus* antibodies might be useful to distinguish mpox from differential diagnoses such as herpes virus infections. In the ulcerative stages, the identified occlusive vasculopathic changes could be an explanation for the severe pain of the lesions reported by some patients. Despite the small number of samples examined, our analysis suggests that the histological findings of mpox are highly dependent on the stage of the biopsied lesion. Therefore, knowledge of all different stages of histology is necessary to reliably diagnose mpox histologically, especially when molecular testing is not available.

## 1. Introduction

Mpox (formerly named monkeypox), a rare viral disease caused by an orthopox virus, was first described in 1970 in the Democratic Republic of Congo [1]. Mpox is the most important human orthopox virus disease since the eradication of smallpox in 1980 and, until recently, was considered endemic to Central and West Africa [2,3,4,5]. During the 2022 outbreak in Europe and North America, numerous cases were reported in countries where the virus is not otherwise endemic [6,7]. In Germany, around 3700 cases have been reported since May 2022 [8]. Until the latest outbreak, animal–human transmission had been considered the major source of infection [9,10,11]. However, during the 2022 outbreak, transmission occurred primarily in the context of sexual activity between men who have sex with men (MSM) [10,12,13,14,15].

According to systematic investigations of the current outbreak in larger patient collectives, the infection shows a staged course: After a short prodromal stage with fever, fatigue, and lymphadenopathy, a few days later, the first skin lesions appear, progressing through various characteristic stages before healing within 14–21 days [2,6,12,13,16,17,18,19,20,21,22,23,24]. Clinically, the lesions usually start with a macule on which a small whitish papule forms centrally. In some cases, secondary pustule formation occurs before the papule ulcerates. The late stage shows ulceration covered by dry necrosis and can be very painful in some cases [6,17,20,22,23,25]. Some lesions heal with scarring [26].

Beyond the typical clinical presentation, the diagnosis of mpox needs to be confirmed by further molecular testing. Currently, the WHO recommends PCR testing as the only method to be used and thus the gold standard. Skin lesion material, including (i) swabs from the lesion surface and/or exudate, (ii) roofs from more than one lesion, or (iii) lesion crusts, is the recommended specimen material to perform appropriate testing. Confirmation of MPXV infection is then based on nucleic acid amplification testing (NAAT) using a real-time or conventional polymerase chain reaction (PCR) to detect unique sequences of viral DNA. Several protocols exist, including two-step protocols in which the first PCR reaction detects only orthopoxvirus (OPXV) but does not identify the exact species. MPXV can then be specifically detected in a subsequent step utilizing PCR or sequencing. Alternatively, generic monkeypox detection (to confirm etiology) can be performed directly, followed by additional PCR testing for specific clade differentiation [27,28,29].

Although not part of the routine WHO diagnostic algorithm, histopathologic and immunohistochemical examinations of skin lesions may also be useful, especially when molecular methods are not available or mpox has not been considered as a differential diagnosis [16,30]. Histopathological features of cutaneous lesions of mpox infections have been described in several case reports and small series [14,25,31,32,33,34]. However, since the publication of the book by Ackermann and Ragaz in 1984, it has been known that inflammatory diseases show different histological patterns at different time points (“the lives of lesions”) [35]. To the best of our knowledge, so far, no histopathologic study of mpox has taken into account the staged course of the disease. Therefore, the focus of this study was to systematically investigate specifically selected skin lesions of mpox in different stages of disease by histology, immunohistochemistry, and electron microscopy.

## 2. Materials and Methods

### 2.1. Sample Collection and Clinical Information

In July 2022, four patients presented to the Department of Dermatology, University Hospital Muenster, Germany, with skin lesions clinically suspicious for mpox. The diagnosis of mpox was confirmed by molecular detection of the virus (PCR) in lesional swabs according to WHO guidelines. As part of the routine diagnostic algorithm in our department, after informed consent, photographs and skin biopsies of representative lesions were taken. Each patient presented with multiple skin lesions in different anatomical regions at different stages of development. Therefore, lesions corresponding to the clinically described stages of mpox were specifically selected for biopsy. In three of the four patients, one biopsy was taken; and in one patient, two skin lesions were biopsied. Retrospectively, the biopsies were assigned to the different clinical stages of mpox. Patients’ characteristics and further clinical information such as time between sexual risk contact, appearance of first skin lesion, and medical consultation as well as symptoms during the course of infection among others were collected retrospectively using the medical records. Detailed information can be found in Table 1. The study was approved by the local ethics committee of the University of Muenster, Germany (Ethik-Kommission Westfalen-Lippe: # 2023-193-f-S; 14 April 2023).

### 2.2. Histology and Immunohistochemistry

Tissue sections (3 µm) were cut from formalin-fixed and paraffin-embedded tissue blocks and stained with hematoxylin and eosin (H&E). For immunohistochemistry, tissue was deparaffinized and rehydrated with distilled water. Subsequently, ‘heat-induced epitope retrieval’ was performed with the respective buffers (CD4, CD8, myeloperoxidase (MPO): EDTA buffer pH 9.1 (DCS, Hamburg, Germany), CD68, and anti-*Vaccinia virus* antibody: citrate buffer pH 6.1 (DCS, Germany)). The following antibodies were used in respective dilutions: CD4 (clone 4B12; Agilent, USA; 1:50 in DCS dilution buffer); CD8 (clone C8/144B; Agilent, Santa Clara, USA; 1:100 in DCS dilution buffer); CD68 (clone KP1; Agilent, Santa Clara, USA; 1:400 in DCS dilution buffer); MPO (polyclonal; Agilent, USA; 1:20,000 in Agilent dilution buffer), and rabbit polyclonal anti-*Vaccinia virus* antibody (clone ab35219, Abcam, 1:200 in DCS dilution buffer). Counterstaining was performed with hematoxylin according to Mayer, and the slides were mounted with Aquatex (Merck, Darmstadt, Germany). Histology was systematically evaluated using a list of criteria for epidermal and dermal inflammatory and vascular changes (Table 2). The involvement of inflammatory cells was assessed on immunohistochemically stained slides (MPO: neutrophils, CD4: T-helper cells, CD8: cytotoxic T cells, and CD68: tissue macrophages). The presence or absence of histopathological and immunohistochemical changes was documented with a semi-quantitative score: o = not present, + = present < 5%, ++ = moderately present between 5% and 50%, +++ = strongly present > 50%, x = not assessable.

### 2.3. Electron Microscopy

Specimens were processed with Karnovsky’s fixative followed by 1% osmium tetroxide, dehydrated, and embedded in EPON resin mixture (Merck, Darmstadt, Germany). Ultra-thin sections were cut with diamond knives, and after mounting on copper grids, the sections were stained with uranyl acetate and lead citrate.

### 2.4. PCR from FFPE Samples

The QIAamp DNA mini kit (Qiagen, Hilden, Germany) was used for DNA extraction from 50 µm sections (5 × 10 µm per biopsy) of formalin-fixed paraffin-embedded tissue. Monkeypox virus (MPXV)-DNA detection was performed with MPXV-specific real-time PCR (LightMix Modular Monkeypox Virus, TIB Molbiol, Berlin, Germany) on a LightCycler 480 II (Roche, Mannheim, Germany) according to the manufacturer’s instructions.

## 3. Results

### 3.1. Clinical Features

All four patients were male and men who have sex with men (MSM). Their mean age was 41 years, ranging from 34 to 47 years. Three of the four patients were HIV-positive, and two had contracted syphilis in the past. The first skin lesions appeared between 12 and 17 days (mean: 14.25; standard deviation: 1.79) after sexual risk contact. The number of lesions varied between a few (three lesions) to multiple (>twenty lesions). Clinically, all patients presented different stages of cutaneous manifestations of mpox: a macular stage (Figure 1a), a papular stage (Figure 1b), a pustular stage (Figure 1c), an early ulcerative/necrotic (Figure 1d), and a late ulcerative/necrotic stage (Figure 1e). All patients’ characteristics and clinical information are summarized in Table 1. Clinical pictures of the different mpox stages are illustrated in Figure 1.
viruses-15-01748-t001_Table 1Table 1Patients’ characteristics and clinical information (STI: sexually transmitted infection, PCR: polymerase chain reaction, FFPE: formalin-fixed paraffin-embedded, Ct: cycle threshold).
Patient #1Patient #2Patient #3Patient #4GenderMMMMAge47344241Sexual orientationHomosexualHomosexualHomosexualBisexualHIV statusNegativePositivePositivePositivePrevious STIsNoneSyphilisNoneSyphilisSymptoms during the course of infectionInguinal lymphadenopathyFever, inguinal lymphadenopathyCephalgia, fatigue, inguinal and submandibular lymphadenopathyFeverAnatomic site of skin lesionsTrunk, lower extremities, genitalsFace, trunk, genitalsFace, trunk, upper extremitiesEnoral, inguinal, genitalsNumber of lesions63>2012Time between sexual risk contact and first skin lesions (days)14121417Time between first skin lesions and medical consultation (days)4347MPXV-PCR from lesional skin swabsPositivePositivePositivePositiveNumber of biopsies1121Anatomic site of biopsyTrunkTrunkTrunkTrunkGenitalsStage of the lesionMaculePapulePustuleEarly ulcerationLate ulceration with dry necrosisMPXV PCR from FFPE biopsies (Ct)37.0119.9922.7023.5421.07


### 3.2. Histopatholgical Features

A total of five biopsies were taken from four different patients representing the different stages of the disease (Table 1). The initial macular stage showed mainly superficial and deep perivascular but also interstitial inflammatory cell infiltrates predominated by lymphocytes (Figure 2a). Some lymphocytes were medium to large in size. The junctional zone showed vacuolar changes (Figure 2b). Only a subtle pallor of the epidermis could focally be identified. Only a few neutrophils were scattered superficially and deeply perivascularly and interstitially within the dermis at this early stage. (Figure 2b). The histology of the papular stage showed a clear increase of inflammatory cells (Figure 2c). Numerous neutrophilic granulocytes were then found in the infiltrate (Figure 2c,d). The epidermis was pale with ballooned keratinocytes. Hair follicle epithelia were similarly affected (Figure 2d). Additionally, some multinucleated keratinocytes and some keratinocytes with eosinophilic inclusion bodies (so-called Guarnieri’s inclusion bodies) could be identified, the latter especially at the upper layer of the lesional skin (Figure 2d). The pustular stage demonstrated a further increase of neutrophilic granulocytes, some of them forming larger intraepidermal and intrafollicular collections (Figure 2e,f). In the stage of early ulceration, the epidermis was mostly necrotic. At the superficial vascular plexus, a strong diapedesis of inflammatory cells were seen, among them neutrophilic granulocytes also within the walls of small vessels as well as the initial formation of intravascular fibrin thrombi. (Figure 2g,h). Histopathologic changes of the vessels themselves, such as fibrin deposition within the vessel walls and perivascular leukocytoclasia, could not be observed at this stage (Figure 2h). These changes could be detected in the late stage of ulceration, in which numerous damaged vessels showed fibrin deposition within the vessel walls as well as intraluminal fibrin thrombi surrounded by erythrocyte sludge together with a strong diapedesis of neutrophils with extravasated erythrocytes and moderate leukocytoclasia around the postcapillary venules (Figure 2j). Details of the systematic semiquantitative histomorphological evaluation of the biopsies are shown in Table 2.

### 3.3. Immunohistochemical Features

Anti-*Vaccinia virus* antibodies marked the cytoplasm of keratinocytes in all samples except the early macular stage (Figure 3a–j). The reaction was particularly strong within the epithelial keratinocytes of the hair follicles and the adjacent epidermis (Figure 3c,e,g). Macrophages surrounding an obliterated blood vessel in the late stage of ulceration were also labelled by the anti-*Vaccinia virus* antibodies (Figure 3i,j). Immunohistochemical staining with myeloperoxidase (MPO) confirmed the increasing invasion of neutrophilic granulocytes from only a few scattered dermal neutrophils in the macular stage (Figure 4a) to extensive infiltration of the epidermis and hair follicles (papular stage, Figure 4b), resulting in a formation of intraepidermal and intrafollicular aggregations (pustular stage, Figure 4c). In the early and especially late stage of ulceration, staining with MPO additionally marked an increasing diapedesis of neutrophils at the postcapillary venules (Figure 4d,e). The inflammatory infiltrate was predominantly composed of CD4+ T cells in the macular stage. In the later stages, the infiltrate was dominated by CD8+ T cells and neutrophilic granulocytes (Figure 4f–o). The immunohistochemical staining with CD68 showed strong reactivity with the macrophages around vessels and adnexal structures in all stages. Additionally, it marked some interstitial macrophages but also showed cross-reactivity with neutrophilic granulocytes (Figure 4p–t). Details of the systematic semiquantitative immunohistochemical evaluation of the biopsies are shown in Table 2.
viruses-15-01748-t002_Table 2Table 2Histomorphological and immunohistochemical assessment.
Biopsy #1Biopsy #2Biopsy #3Biopsy #4Biopsy #5Clinical StageMaculePapulePustuleEarly ulcerationLate ulcerationMain inflammatory pattern Superficial, deep perivascular, and interstitial Superficial, deep perivascular, and interstitial Superficial, deep perivascular, and interstitial Superficial, deep perivascular, and interstitial Superficial, deep perivascular, and interstitialEpidermal changesParakeratosisooo+++Spongiosis++++++Pallor of epidermis+++++++++Necrotic keratinocyteso++++++++AcantholysisoooooBallooning+++++++++Reticular degenerationo+oooMultinuclear keratinocyteso+++++Vacuolization of the junctional zone++++++++++Guarnieri bodieso+++oVascular/perivascular changesCapillary/postcapillary venules thrombosisooo++++Neutrophilic infiltration of small vessel wallsooo+++Fibrin within vessel wallsoooo++Fibrin perivascularoooo++Perivascular leucocytoclasiaoooo++Extravasated erythrocytes++++++++ImmunohistochemistryAnti-*Vaccinia virus* antibodyEpidermalo++++++++++++Follicularo++++++xxNeutrophils (MPO)Intraepidermal/intrafollicular+++++++++++Intravascular++++++++Perivascular++++++++Small vessel wallsooo+++Diapedesis+++++++Interstitial+++++++++T-cell infiltrate (CD3) and subsets (CD4 and CD8)Perivascular+++++++++Periadnexal++++++Interstitial+++++Epidermotropism/Adnexotropism+++++T-helper cells (CD4)++++++++Cytotoxic T cells (CD8)+++++++++CD4/CD8-ratio10:11:11:41:41:3Macrophages (CD68)Perivascular/periadnexal++++++++++++Interstitial+++++Semi quantitative score: o = not present, + = present, ++ = moderately present, +++ = strongly present, x = not assessable.


### 3.4. Electron Microscopic Features

Additionally, two of the five samples (pustule stage and late ulceration stage) were analyzed with electron microscopy, allowing for visualization of typical ovoid-shaped viral particles with a biconcave outline of the core within the cytoplasm of epidermal keratinocytes and hair follicle epithelia (Figure 5). 

### 3.5. PCR from FFPE Samples

In all five skin biopsies, MPXV-DNA was detectable by PCR using the above-mentioned protocol. The respective Ct values can be found in Table 1.

## 4. Discussion

The current outbreak of mpox beginning in May 2022 has increasingly drawn the attention of medical professionals worldwide to this previously rare viral zoonosis [6]. 

According to the current studies, MPXV seems to infect the host first via the mucous membranes and is then transported to the regional lymph nodes by Langerhans cells [36]. After a short prodromal stage with viral replication in the lymphoid organs, the generalized seeding of the viruses occurs via the blood and, subsequently, characteristic skin lesions typically develop only in circumscribed areas on the skin [18]. The individually localized lesions regularly pass through five characteristic stages (macule, papule, pustule, and early and late ulceration), which could also be detected in our collective.

The histology of mpox lesions has been described by different groups before, all of them observing similar findings [13,14,25,31]. Characteristic changes were mostly described at the epithelium: (i) acanthosis and pallor of the epidermis, (ii) ballooning of keratinocytes and multinucleation, (iii) eosinophilic intracytoplasmic inclusion bodies (so-called Guarnieri bodies), and (iv) so-called exocytosis of neutrophils. Our analysis confirms these findings in the papular, pustular, and ulcerative stages but also shows that histology changes significantly from stage to stage. We further add findings in the early macular stage, which, to the best of our knowledge, has not been described in previous studies. In addition, our study is the first to put the staged progression of mpox skin lesions into an orderly sequence.

In the literature, it is frequently mentioned that the histology of mpox can be confused with herpes virus infections due to similar pallor of the epidermis with the ballooning and formation of multinucleated keratinocytes [14,31,37]. However, in herpes virus infections, histologically, a “steel gray” appearance of the nuclei with margination of the chromatin is typical (Appendix A). In orthopoxvirus infections, eosinophilic inclusion bodies are detectable in the cytoplasm of the infected keratinocytes [31,36]. Moreover, we could not observe acantholysis in our specimens, which is a common finding in herpes infections [37]. Nevertheless, especially in advanced stages, differentiation can still be difficult, and therefore, immunohistochemical staining with the anti-*Vaccinia virus* antibodies can be useful. Our immunohistochemical analysis could confirm the results of previous studies showing that this antibody is suitable to detect mpox on histological material with strong reactivity in affected keratinocytes [14,31]. This, however, does not apply to the early macular stage in which the immunohistochemistry with anti-*Vaccina virus* antibodies remained negative. In this stage, virus production is probably still below the detection limit of the antibody, as MPXV-DNA was already found by real-time PCR in the macular lesion. Hence, the early macular stage is associated with most differential diagnostic difficulties. Showing superficial and deep perivascular lymphocytic infiltrates, vacuolar alterations along the junctional zone, and only subtle keratinopathic changes, the histology of mpox at this early stage can easily be mistaken for drug reactions, cutaneous lupus erythematosus, pityriasis lichenoides, or cutaneous T-cell lymphoma. An accurate diagnosis of mpox at this very early stage therefore only seems to be possible with additional methods such as PCR.

Notably, immunohistochemical staining with the anti-*Vaccinia virus* antibodies detected viral proteins not only in the keratinocytes of the interfollicular epidermis but also within hair follicle epithelia. This was also confirmed by electron microscopy and is in line with previous studies [14,31]. Of particular interest is the strong infestation of hair follicle epithelia, which has already been demonstrated in other human studies [14,31] but also in a study with monkeys [4]. Follicle-centered viral infections are known, among others, from herpes viruses and, here especially, *Varicella zoster virus* (VZV) [38,39]. In the latter cases, it is assumed that VZV is transported from the dorsal root or trigeminal ganglia via myelinated nerves which terminate at the isthmus of hair follicles. To date, there is no evidence that this also might be the case in mpox. Nevertheless, there are reports on postvaccinal folliculitis after administration of the smallpox vaccine, showing deposition of vaccinia virus (an orthopox virus similar to MPXV) proteins within epithelial cells of the folliculosebaceous units, suggesting that epithelial cells might have some sort of susceptibility for orthopox viruses [40,41]. On the other hand, it could be hypothesized that the hair follicles may be some sort of “locus minoris resistenciae”, in which the viruses penetrate more easily into the skin, e.g., by an additional smear infection during the initial contact. In the course of the infection, the T-cell response is then primarily triggered around the hair follicle as it is known to occur, e.g., in herpes folliculitis [39,42]. This could explain the occurrence of usually only few cutaneous lesions in mpox patients. The strong immune response around hair follicles also appears to contribute to the papulo-ulcerative efflorescence of cutaneous mpox lesions.

We are well aware that due to the small number of specimens that we have examined, only limited conclusions can be drawn. Nevertheless, our analysis of the inflammatory patterns, the composition of the infiltrate, and the subsequent vascular changes suggest the following immunological response: (i) The lesions are primarily triggered by a T-cell response dominated by CD4+ T-helper cells, probably directed against the infected keratinocytes. The skin lesions develop with a time delay because T-cell priming in the lymph node is probably required first. (ii) The subsequent cytotoxic CD8+ T-cell response results in the ballooning degeneration and necrosis of the infected keratinocytes, a known trigger for the recruitment of neutrophil granulocytes, which dominate in the lesions as they progress. (iii) The massive diapedesis of neutrophilic granulocytes from the postcapillary venules then probably leads to the vascular closures of the superficial vessels, which exacerbate to ulcerations that we observed in the late stages. This phenomenon is called immunothrombosis and has been described in other viral diseases, for example, in COVID-19 [43,44,45]. We have recently also observed this mechanism in the treatment of ano-genital warts with ingenol mebutate [46]. Obliterating the vascular changes typically causes a strong ischemic pain and therefore might be the histopathologic correlate for the sometimes-massive pain that mpox patients report. It can therefore be assumed that both direct viral cytopathic effects and the immunological response of the host contribute equally to the clinically as well as the histologically stage-like appearance of mpox.

Due to the retrospective nature and the small number of cases, our statements and conclusions are not generalizable. The largest study population examined histologically was described in Spain and included 20 patients [14]. However, the early macular stage was not described there. Moreover, all samples in this study were collected from the same hospital and processed in the same laboratory, providing comparability within the material studied. Nevertheless, it would be desirable in the future to study a larger number of samples including all stages of mpox lesions, also from the same patient, to verify our results.

To summarize, the histological changes of mpox are highly dependent on the stage of the clinical lesions. Especially, the early macular stage can be easily missed histologically. Therefore, clinicopathologic correlations and additional virologic testing with PCR as the diagnostic gold standard are required to establish the diagnosis of mpox with certainty. Since the further spread of mpox cannot be predicted and is still ongoing at present, mpox has to be included in the differential diagnostic repertoire. Knowledge of the histological changes is helpful in diagnostics, especially when molecular methods are not available.

## Figures and Tables

**Figure 1 viruses-15-01748-f001:**
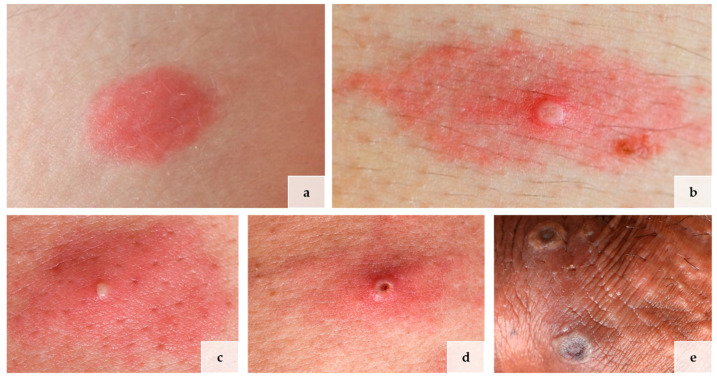
Lives of lesions in mpox in 4 different patients: (**a**) macular stage, (**b**) papular stage, (**c**) pustular stage, (**d**) early and (**e**) late ulceration. Picture (**c**,**d**) are from the same patient.

**Figure 2 viruses-15-01748-f002:**
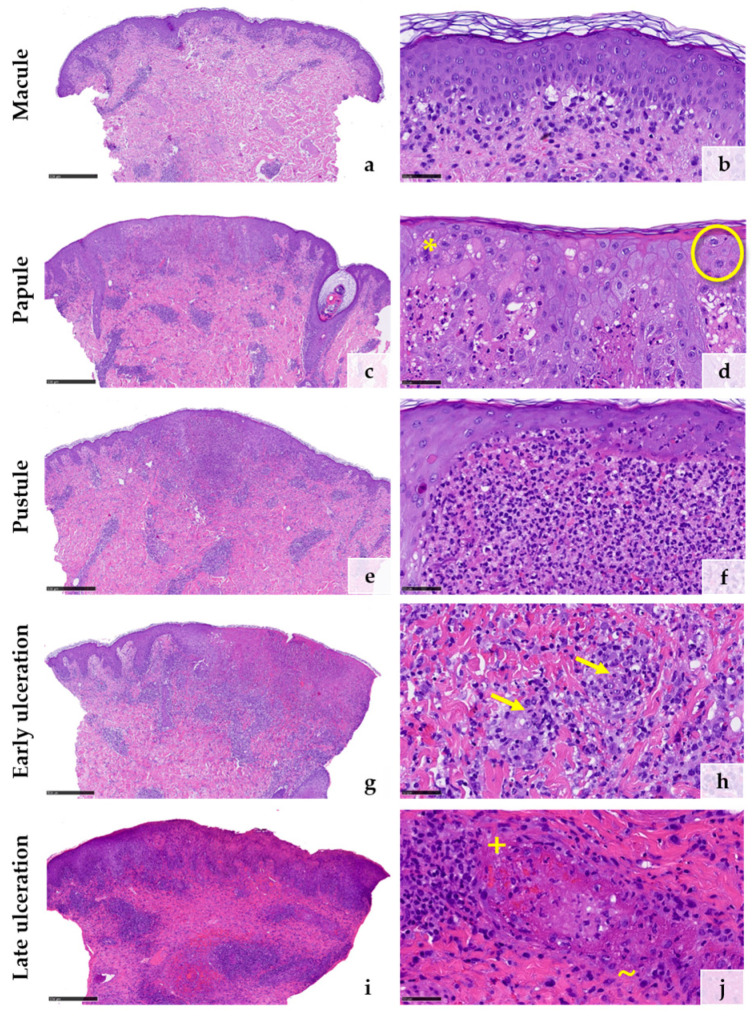
Histological features corresponding to the clinical stages of mpox lesions from macule (**a**,**b**) to papule (**c**,**d**), pustule (**e**,**f**), early ulceration (**g**,**h**), and late ulceration (**i**,**j**) ((**d**), * = multinucleated keratinocytes; O = Guarnieri’s inclusion bodies; (**h**) → = neutrophil granulocytes within the vessel walls; (**j**), + = fibrinoid deposition within the vessel walls; ~ = leukocytoclasia). (Stain: hematoxylin and eosin, scale bar 500 µm (**a**,**c**,**e**,**g**,**i**), scale bar 50 µm (**b**,**d**,**f**,**h**,**j**)).

**Figure 3 viruses-15-01748-f003:**
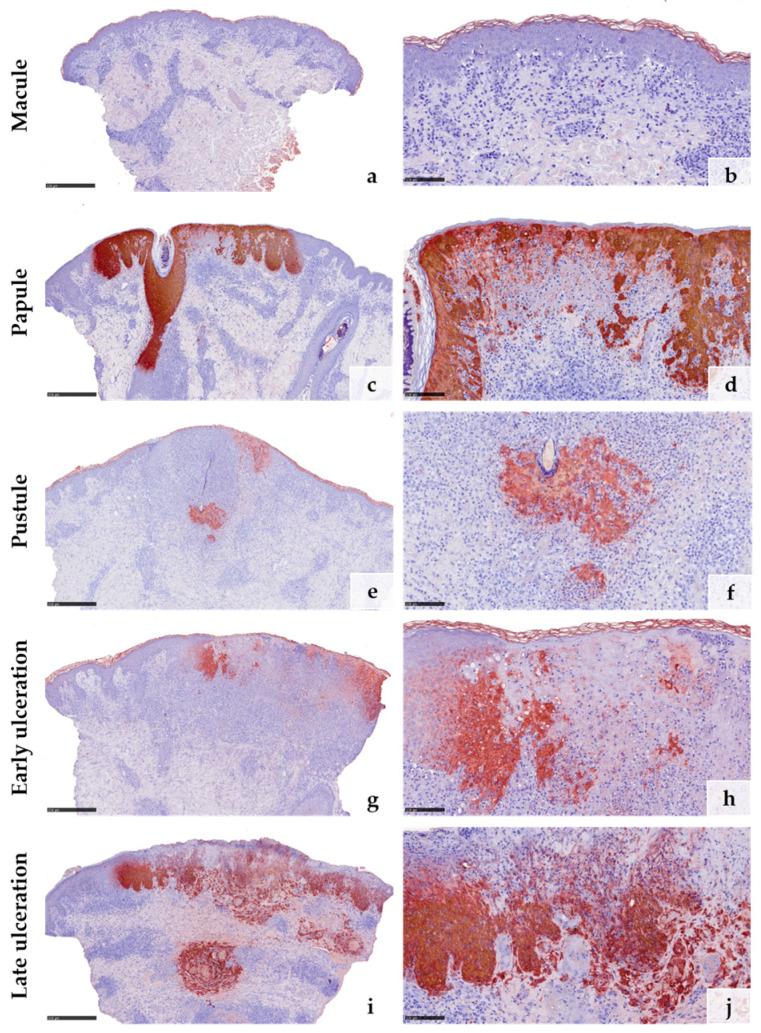
Immunohistochemical reaction to anti-*Vaccinia virus* antibodies in the different clinical stages of mpox: No reaction is detected in the early macular stage (**a**,**b**). In all the other stages (**c**–**j**), the reaction is strong, particularly within the epithelial cells of the hair follicles (**c**,**e**,**g**). Some macrophages show cross-reactivity with anti-*Vaccinia virus* antibodies (**i**,**j**). (Stain: hematoxylin and eosin, scale bar 500 µm (**a**,**c**,**e**,**g**,**i**), scale bar 100 µm (**b**,**d**,**f**,**h**,**j**)).

**Figure 4 viruses-15-01748-f004:**
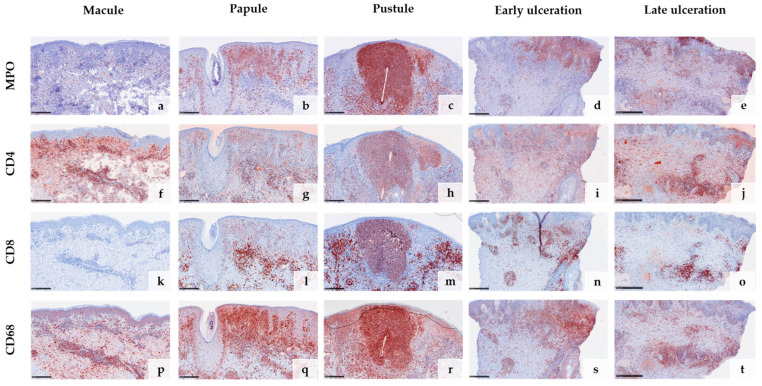
Immunohistochemical analysis of the inflammatory infiltrate in different stages of mpox skin lesions: Staining with myeloperoxidase (MPO) confirms the increasing invasion of neutrophilic granulocytes from early to late stages (**a**–**e**) and marks the strong diapedesis of neutrophils at postcapillary venules in the stages of ulceration (**d**,**e**). In the early stage, the inflammatory infiltrate is predominantly composed of CD4+ T cells (**f,k**), shifting towards CD8+ T cells and neutrophils in the later stages (**g**–**j**,**l**–**o**). CD68 shows strong reactivity with macrophages around vessels and adnexal structures but also marks some cross-reactive neutrophils (**p**–**t**). (Scale bar 250 µm).

**Figure 5 viruses-15-01748-f005:**
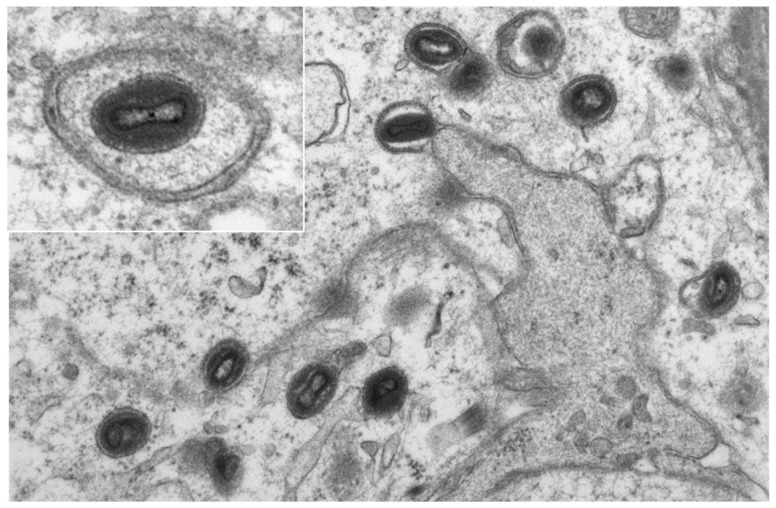
Electron-microscopic picture of mpox in keratinocytes of the epidermis from the biopsy of patient 5. (Original magnification × 10,000, insert × 20,000.)

## Data Availability

The authors confirm that the data supporting the findings of this study are available within the article. Raw data are available on request from the corresponding author, P.S.

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
