# Peer review of "Lives of Skin Lesions in Monkeypox: Histomorphological, Immunohistochemical, and Clinical Correlations in a Small Case Series"

_viruses, 2023, doi:10.3390/v15081748_

Round 1

Reviewer 1 Report

I thank the authors for the concise manuscript about the lives of skin lesions in mpox cases. I have some small recommendations.

Title: is "immuhistochemical" the right spelling?

In the abstract in line 24 you write: "... that histology of mpox is highly dependent on the stage..." do you mean the success or the interpretation of the histology?

in Section 2.1. it might be helpful for the reader to have more detailed information on sample collection here rather than later in the results section. Like at what time point after infection/contact did the patients present themselves? How many samples per patient were taken/analyzed? I mistakenly thought you took follow up samples from the same patients. Maybe you could refer to table 1 and add missing information there if you don't want to add everything in the main text.

Table 1: the described symptoms, are that symptoms at time point of presentation of patient or during the course of infection? Did the patients have different stages of lesions on the different anatomic sites? please clarify

Section 3

there is no section 3.2. but 3.5.  appears twice

Figure 1: are d) and e) from the same patient, different location of lesion? Also I recommend to have a more detailed figure description so the reader doesn't have to go back to the text.

Figure 3 & 4: A more detailed figure description might be helpful to understand what the reader can see.

section 3.5. Why did you do EM only for 2 of 5 samples? And is it really relevant in the context of histology and immunochemistry in this manuscript?

section 3.5. PCR: Since you have only one sample per stage I would be careful with the statement about the peak of viral load in the papular stage. Detected viral load greatly depends on sample quality, how good the sample was taken, the volume it was resuspended in etc.. It is difficult to compare viral loads in these kind of samples, best would be to include a reference gene for comparision. Also crusts are usually loaded with virus.

Section 4 line 235 you write "... which could also be detected in our collective in the corresponding order." What do you mean by corresponding order?

line 244: maybe highlight 1 or 2 added findings?

As you mention a lot the comparison to herpes virus infections, it might be helpful to include a picture of a herpes virus histology for direct visual comparison?

line 282: you write "... orthopox virus similar to mpox", mpox is the disease, rather write MPXV

line 316-317: it might be desirable to include all stages of mpox lesions from the same patient

Reviewer 2 Report

The manuscript “Lives of skin lesions: Histomorphological, immuhistochemical and clinical correlation of monkeypox” is well written and reports nice pictures.

However, some improvements should be done ang generally it doesn’t bring any novelty in the specific topic.

The manuscript reports the description of the skin lesions of four cases: the title should be modified accordingly to avoid misunderstanding to the reader that now expects to read a review or a description of high number of cases and this is not the case. Please, use a title just reporting that the skin lesions from four cases will be described.

In the text the reader often has the suggestion that virological diagnosis is not required: the appropriate algorithm accordingly to WHO guidelines should be described. If it was not done for the cases described, it should be explained and discussed. It seems that the diagnostic algorithm in the practice was that reported by the authors but it not true: please, clarify that it is a study of skin lesions of four cases.

The virological diagnosis must include Transmission electron microscopy (performed by a virologist) of the skin lesion biopsy (very small fragment are enough) and serology to search for the human poxvirus immunity. Cultivation, as it was mentioned (line 52) is not recommended as this virus is a BLS 4 agents and the manipulation must be done in appropriate safety condition. The cited references (27 and 28) are not the guidelines for the diagnosis, but the description of an epidemic in high-risk area where diagnostic tools were used to confirm the cases for “case definition” standing the unusual human infection with monkey pox and the years (2017). Please, check that references are cited appropriately.

Was hypothesized the way of infection in all the 4 cases? were partners also infected?

The manuscript should be reduced to a short communication/case reports as it adds novelty on the topic.

The figures albeit very nice, albeit not “new” in the scientific community, should be reduced to the reduction of the format.

some typing errors.

Round 2

Reviewer 2 Report

Thank you to have revised the manuscript; in the revise version it was improved significantly.